# Accounting for the climate benefit of temporary carbon storage in nature

H. Damon Matthews [1] ✉, Kirsten Zickfeld [2], Alexander Koch [2,4] & Amy Luers[3]

Nature-based climate solutions can contribute to climate mitigation, but the vulnerability of land carbon to disturbances means that efforts to slow or reverse land carbon loss could result in only temporary storage. The challenge of accounting for temporary storage is a key barrier to the implementation of nature-based climate mitigation strategies. Here we offer a solution to this challenge using tonne-year accounting, which integrates the amount of carbon over the time that it remains in storage. We show that tonne-years of carbon storage are proportional to degree-years of avoided warming, and that a physically based tonne-year accounting metric could effectively quantify and track the climate benefit of temporary carbon storage. If the world can sustain an increasing number of tonne-years alongside rapid fossil fuel $CO_2$ emissions reductions, then the resulting carbon storage (even if only temporary) would have considerable and lasting climate value by lowering the global temperature peak.

Nature-based climate solutions (NbCS) represent a suite of actions that have the potential to contribute to climate mitigation goals by enhancing the sequestration of carbon through protection, restoration or improved management of natural lands[1-3]. Examples of NbCS strategies to slow or reverse the loss of carbon from land ecosystems include avoided or delayed deforestation, reforestation, and enhanced soil carbon sequestration. NbCS efforts have received criticism or scepticism related to their ineffectiveness in increasing carbon sequestration, their emphasis on measuring carbon flows rather than carbon stocks, their limited consideration of non-carbon climate effects such as those arising from land surface changes, or their singular focus on carbon at the expense of other environmental or social values[4-10]. However, if well implemented, many NbCS have the potential to generate measurable carbon gains while also providing other environmental and social benefits such as enhanced biodiversity, air and water quality, livelihoods and culture[1,2,9,11,12].

A key challenge of land-based NbCS relates to the longevity of stored carbon in ecosystems that are vulnerable to both natural and human-driven disturbances[13,14]. This is particularly relevant for strategies such as avoided deforestation, reforestation and afforestation, which aim to preserve or increase carbon storage in above-ground biomass, though impermanence can also be of concern for strategies

that seek to increase soil carbon storage. Climate-driven increases in wildfires, droughts and insect outbreaks have the potential to decrease the turnover time of forest carbon stocks, leading to an increased likelihood of carbon loss at many locations[13]. Furthermore, anthropogenic activities could also lead to the loss of carbon that was previously sequestered[15], for example if deforestation activities are only delayed rather than being avoided in perpetuity or if previous land management practices are reversed. Both sets of processes have the potential to affect the permanence of land-based carbon storage, leading to the possibility of only temporary carbon storage resulting from any particular carbon sequestration effort.

At large spatial and temporal scales, only a small portion of land carbon is lost as a result of natural disturbances, and this carbon loss is generally balanced by regrowth from previously disturbed areas[16]. From this perspective, any effort to preserve or increase land carbon storage will have a positive climate effect regardless of whether the carbon remains stored temporarily or permanently. However, from a carbon accounting perspective, the longevity of individual units of stored carbon is a critical uncertainty, especially in the context of increasing frequency of natural disturbances[13]. Corporate and other sub-national (e.g. municipal) carbon accounting activities often include carbon storage in natural systems as part of their carbon

[1]Concordia University, Montreal, QC, Canada. [2]Simon Fraser University, Vancouver, BC, Canada. [3]Microsoft Corporation, Seattle, WA, USA. [4]Present address: Trove Research, Harpenden, UK. ✉e-mail: damon.matthews@concordia.ca

ledgers, and many corporations are explicitly looking to nature-based carbon removal as a strategy to achieve their net-zero emissions targets[17]. In this context, the impermanence of land-based carbon storage can be highly problematic, especially if enhanced land-based carbon storage is seen as a viable alternative to an equivalent amount of avoided fossil fuel $CO_2$ emissions. Given that the climate effect of fossil fuel $CO_2$ emissions is effectively indefinite[18–20], replacing a unit reduction of $CO_2$ emissions with a unit increase of temporary land carbon storage would lead to increased total emissions over time and consequently more long-term warming[21–23].

The challenge of accounting for temporary carbon storage in relation to fossil fuel emissions has led some researchers to propose the idea of tonne-year accounting, where one tonne-year is defined as one tonne of carbon that is stored in some carbon reservoir for one year[24,25]. Tonne-year accounting can be used to measure the time-integrated amount of carbon that is stored in temporary land carbon stocks by multiplying the amount of stored carbon by the amount of time in remains stored[24]. Similarly, tonne-year accounting can be used to estimate the atmospheric $CO_2$ response to an emission by integrating the amount of carbon remaining in the atmosphere over some chosen time horizon[25,26]. To quantify the value of temporary carbon storage, tonne-year accounting is generally used to calculate a cost-benefit ratio to compare the cost of an emission to the benefit of delaying that emission for some period of time. These analyses lead to proposed equivalency factors that aim to represent how many tonne-years of temporary storage are required to have equal value as one tonne of permanent storage[24–26]. Such equivalency factors range from about 30 to 130, and have been promoted as the basis for carbon offsets, whereby a given number of temporary tonne-years (for example from delayed deforestation of some area of land) are presented as equivalent to a unit emission of fossil fuel $CO_2$[24–26].

This form of tonne-year accounting has been critiqued from several angles, including its use of subjective economic discount rates and arbitrary time-horizon choices to evaluate the cost of an emission and the benefit of temporary storage leading to a delayed emission[27,28]. Tonne-year accounting also has weak grounding in the actual climate response to emissions and removals of $CO_2$. Previous analyses have focussed only on the atmospheric $CO_2$ response to a pulse $CO_2$ emission, and have not quantified the temperature response to this atmospheric $CO_2$ change, nor considered the difference between short- and long-term climate responses[27,28]. Proponents of tonne-year accounting have consequently proposed equivalency factors that have little bearing on the climate consequence of temporary vs permanent storage[29,30]. As a result, tonne-years have not been widely embraced as a metric to represent the climate effect of temporary carbon storage resulting from NbCS efforts.

Here, we offer a solution to the challenge of accounting for temporary carbon storage in nature as a contribution to national or global climate mitigation goals, via a revised approach to tonne-year accounting. We propose that tonne-years, if applied only as a physical metric of carbon storage over time rather than an economic metric of equivalency, could be used effectively to estimate the climate response to nature-based carbon storage. Using an intermediate-complexity global climate model (see Methods), we show that tonne-years of temporary carbon storage do have an important and quantifiable climate effect that emerges from the well-understood climate response to cumulative $CO_2$ emissions. On this basis, we argue that tonne-year accounting could be reimagined, not as an offset metric, but rather as an independent tracking and reporting metric for nature-based carbon removals and avoided emissions that would not require the permanence of carbon storage to be demonstrated a priori. Such an approach to tonne-year accounting could mobilise the potential of temporary carbon storage as an effective climate mitigation action.

## Results and discussion
### Tonnes per year, tonnes, and tonne-years
The simulations presented here represent the effect of avoided land-based $CO_2$ emission and/or removals as an annual decrease of pre-scribed emissions relative to the land-use emissions of the baseline scenario (see "Methods"). We decreased emissions beginning in the year 2022 by 3 $GtCO_2$ per year, reflecting the global cost-effective mitigation potential of actions such as avoided deforestation and reforestation[1]. In the permanent carbon storage simulations, we maintained the decrease in land-use emissions until either the year 2050 or throughout the simulation (Fig. 1a, solid lines). Temporary storage simulations followed the same initial decrease in emissions until either the year 2037 or 2050 (Fig. 1a, dashed lines). Rather than returning to the baseline emission level (as in the 2050 permanent storage case) emissions subsequently increased relative to the baseline scenario such that all carbon that was previously stored was re-emitted to the atmosphere over a period of either 15 or 50 years (Fig. 1a; dashed lines). These temporary storage simulations could reflect scenarios whereby deforestation activities are delayed rather than avoided permanently, or whereby sequestered carbon is subsequently lost as a result of climate-driven increases in natural disturbances. When plotted in terms of cumulative emissions, permanent storage resulted in lower cumulative land-based $CO_2$ emissions throughout the simulation, whereas temporary storage resulted in lower cumulative emissions for the period with increased storage, followed by a return to the baseline scenario level of cumulative emissions when all stored carbon was re-emitted (Fig. 1b).

Here, we represent tonne-years of carbon storage as a running total of the amount of stored carbon multiplied by the time over which it remained stored, which we calculated as the time-integral of the cumulative emissions difference from the baseline scenario (shaded region in Fig. 1b). Regardless of whether carbon storage was temporary or permanent, the total number of tonne-years initially increased with time (Fig. 2a). During the period that land carbon emissions were lower than the baseline simulation, tonne-years accumulated at an increasing rate, since each additional removal or avoided emission added to the quantity of carbon that was integrated over the time it remained stored. For the permanent storage scenarios, when emissions returned to the baseline scenario level, the number of tonne-years continued to accumulate at a constant rate as the total amount of stored carbon was added to the tonne-year total for each additional year that it remained stored. For the temporary storage scenarios, however, when the previously stored carbon began to be re-emitted, the accumulation of tonne-years slowed in proportion to the amount of stored carbon that was lost. When all stored carbon had been re-emitted, tonne-years stopped accumulating and subsequently remained constant for the remainder of the simulation (Fig. 2a).

### Climate response to tonne-years of temporary carbon storage
In response to fossil fuel $CO_2$ emissions and other climate forcings from the two most ambitious mitigation scenarios amongst the group of SSP marker emission scenarios (SSP1-1.9 and SSP1-2.6), global temperatures in the baseline land-use configuration peaked just below 1.5 °C for SSP1-1.9 and just above 1.7 °C for SSP1-2.6 (Fig. 2c). The timing of this temperature peak depended on the timing of net zero fossil fuel $CO_2$ emissions in these scenarios, whereby the slower transition to net zero in SSP1-2.6 led to a temperature peak occurring about 30 years later than in SSP1-1.9.

Decreased land $CO_2$ emissions in these scenarios relative to the baseline land-use scenario lowered global temperatures in proportion to the cumulative emissions difference over time. For the temporary storage scenarios, the temperature decrease reached a maximum of 0.02 and 0.05 °C below the baseline temperatures, and then returned to the baseline temperatures following re-emission of the stored

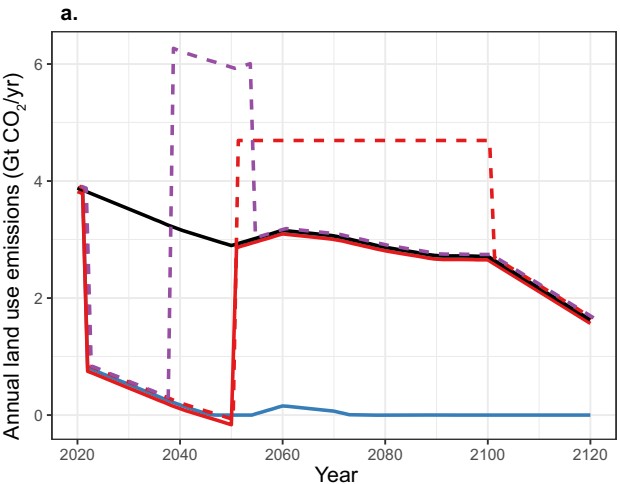

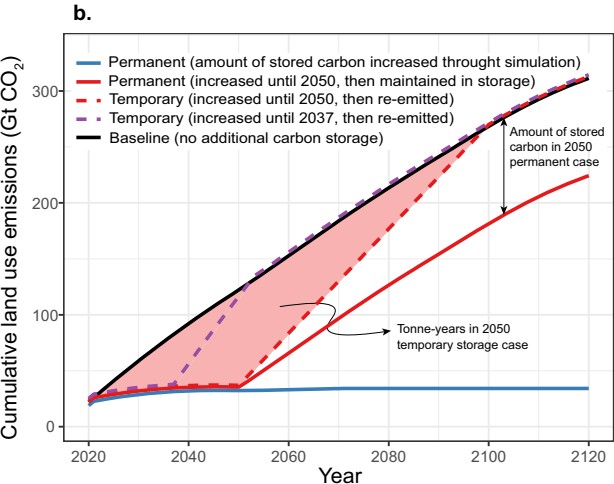

**Fig. 1 | Prescribed land-use CO₂ emission scenarios. a** Annual land-use CO₂ emissions, representing the net land carbon flux for the baseline scenario (black line), two permanent storage scenarios (solid red and blue lines) and two temporary storage scenarios (dashed purple and red lines). Permanent storage simulations were represented by decreased emissions from the baseline, sustained until 2050 (solid red line) or until the end of the simulation (solid blue line). These permanent storage simulations represent cases where decreased net land-use emissions lead to increased land carbon storage that remains sequestered throughout the simulation. Temporary storage simulation followed the same initial decrease until either 2037 (purple dashed line) or 2050 (dashed red line), after which emissions were abruptly increased relative to the baseline to represent a switch from carbon sequestration to re-emission of previously stored carbon.

These temporary storage simulations represent a case where carbon is sequestered until 2037 or 2050, and then gradually re-emitted to the atmosphere over a period of either 15 years (dashed purple) or 50 years (dashed red). **b** Cumulative land-use CO₂ emissions, plotted relative to the year 2015, where lower cumulative emissions represent either avoided land-use emissions or land-based carbon removal relative to the baseline land-use emission scenario. For the temporary storage simulations, the cumulative land-use emissions returned to the baseline level after all stored carbon was re-emitted, whereas permanent storage resulted in lower cumulative land-use CO₂ emissions throughout the simulation. The shaded area in (**b**) illustrates the total number of tonne-years of storage achieved in the 2050 temporary storage scenario. Note that in both panels, overlapping lines are offset slightly for improved clarity. Source data are provided as a Source data file.

carbon (Fig. 2b, c). In the case of the earlier temperature peak of the SSP1-1.9 scenario, both temporary storage scenarios led to a lower peak temperature. In the later peak temperature scenario (SSP1-2.6), peak temperature was only decreased in the case that some amount of temporary storage was sustained until the second half of the century. For the permanent storage simulations, the temperatures continued to decrease relative to the baseline for as long as emissions remained lower than the baseline emissions; when emissions returned to the baseline level, the temperature difference was sustained for as long as the carbon remained stored. The effect of our land carbon storage scenarios did not depend on the background emission scenario, with both SSP1-1.9 and SSP1-2.6 showing the same temperature difference resulting from both temporary and permanent land storage (scenario SSP1-1.9 is plotted in Fig. 2b).

From these simulations, we can identify a clear relationship between tonne-years of land carbon storage (Fig. 2a) and the simulated temperature difference (Fig. 2b). During the period of time that tonne-years accumulated faster than linearly, global temperatures diverged from the baseline temperatures. A constant rate of accumulation of tonne-years resulted in a global temperature difference that remained stable over time. However, a decreasing rate of tonne-year accumulation resulted in a decreasing temperature difference, with temperatures returning to the baseline at the time that total tonne-years stopped accumulating.

These results suggest that tonne-year accounting could be used to track the effect of stored land carbon over time, and to infer the temperature benefit of that storage as a function of the rate of increase of the total number of tonne-years in the system. A constant rate of tonne-year accumulation would result in a sustained temperature benefit. An increasing rate of tonne-year accumulation would result in an increasing temperature benefit over time. A decreasing rate of accumulation would result in the erosion of previously accrued temperature benefit. Constant tonne-years over time would mean that all

previous temperature benefit has been lost and global temperature will have returned to where it would have been in the absence of any temporary carbon storage.

Importantly then, the potential for temporary land carbon storage to lower peak warming can also be tracked as a function of the rate of accumulation of total tonne-years. If the total number of tonne-years increases linearly or faster than linearly until the time that fossil fuel CO₂ emissions reach net zero, then the resulting temporary land carbon storage would lower the peak temperature that is reached. However, if the rate of tonne-year accumulation slows and decreases to zero before net-zero fossil fuel CO₂ emissions are achieved, this temporary land carbon storage would have no effect on peak warming (Fig. 2).

## What is the climate equivalent of a tonne-year?
The proportionality of cumulative emissions to global temperature change is well represented by the Transient Climate Response to cumulative CO₂ Emissions (TCRE), where the TCRE is a constant value that approximates the temperature increase caused by total CO₂ emissions over time[21,22]. This TCRE relationship can be applied to the case of carbon removal and storage as follows:

$$\Delta T(t) = \text{TCRE}^*(E(t) - R(t)) \tag{1}$$

Here, $\Delta T(t)$ is the global temperature change over time, $E(t)$ represents the cumulative emissions and $R(t)$ represents the cumulative CO₂ removals achieved over time via land carbon storage. Similarly, the temperature difference from the baseline scenario in our land carbon storage simulations is proportional to the cumulative avoided (removed) land CO₂ emissions:

$$\Delta T_R(t) = \text{TCRE}^* R(t) \tag{2}$$

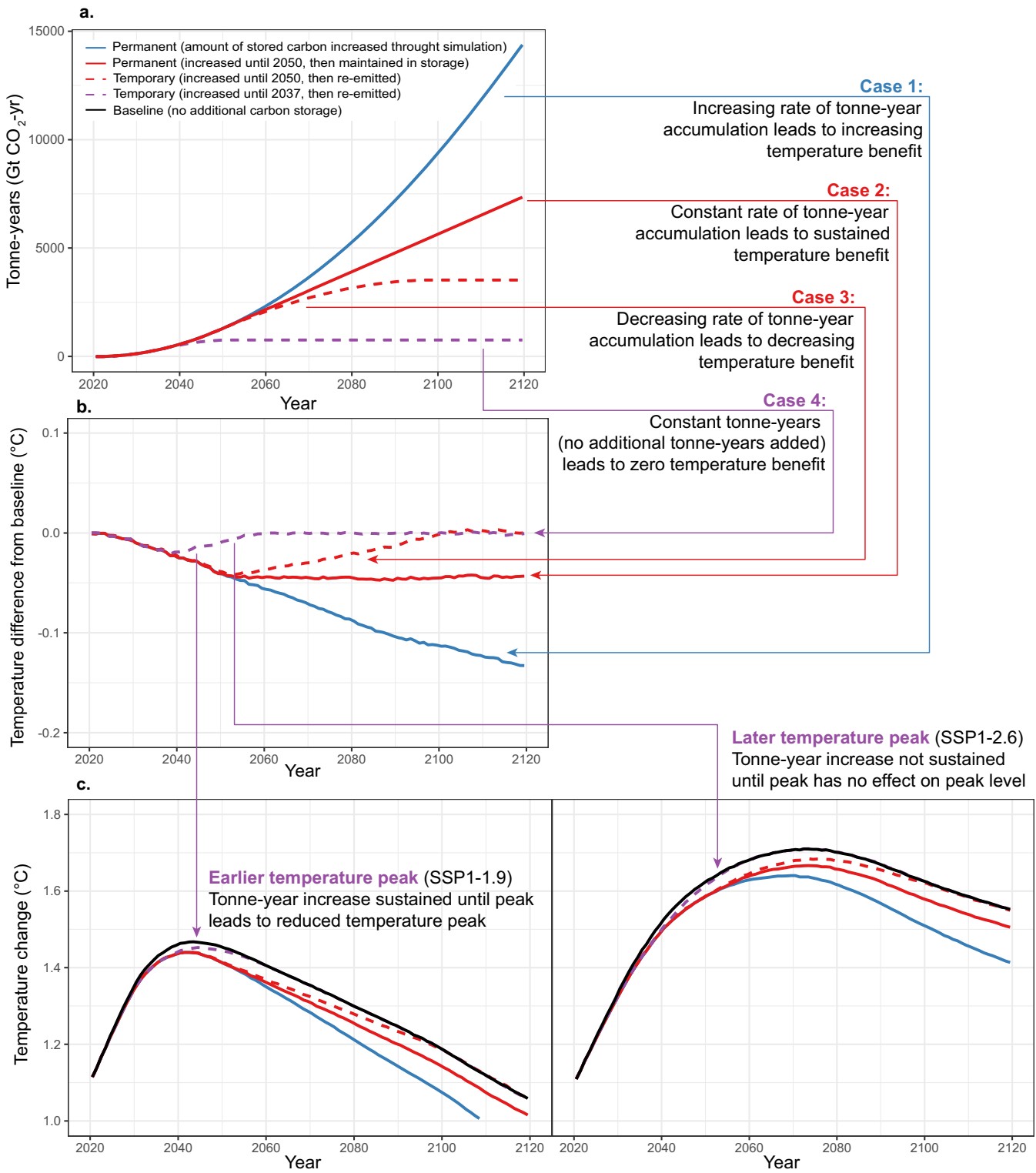

**Fig. 2 | The climate effect of tonne-years of land carbon storage. a**. Tonne-years increased in all simulations, where permanent storage (solid red and blue lines) is characterised by a constant or increasing rate of accumulation of tonne-years. In contrast, temporary storage (dashed red and purple lines) resulted in an increasing rate of accumulation during the time that the amount of stored carbon was increasing, followed by a decreasing rate of accumulation when previously stored carbon began to be re-emitted. When all stored carbon had been lost, the number of tonne-years stopped increasing and remained constant over time. **b**. The global temperature difference from the baseline was proportional to the rate of accumulation of tonne years, such that a constant or increasing rate of tonne-year accumulation resulted in a sustained or increasing temperature difference from the baseline. A decreasing rate of tonne-year accumulation led to the erosion of the temperature difference, followed by temperatures returning to the baseline when tonne-years stopped increasing. **c**. In cases where a constant or increasing rate of tonne-year accumulation was sustained past the point of peak temperature, this resulted in a lower temperature peak. The four cases labelled in the figure highlight representative periods of time along each tonne-year trajectory in which the rate of accumulation of tonne-years increased (Case 1), remained constant (Case 2), decreased (Case 3) or was equal to zero (Case 4). Source data are provided as a Source Data file.

By definition, the tonne-year metric (TY) represents the time-integral of avoided cumulative emissions:

$$TY = \int R(t) \qquad (3)$$

Combining Eqs. (2) and (3), it is apparent that tonne-years are proportional to the time integral of the temperature difference between the temperature curves shown in Fig. 2. We define this time-integrated temperature difference as the "degree-years" (DY) of avoided warming:

$$DY(t) = TCRE*TY(t) \qquad (4)$$

This relationship is illustrated in Fig. 3, which shows that the degree-years in each model scenario are proportional to the number of tonne-years, with the proportionality constant equal to the TCRE of this model (-0.45 degree-years of avoided warming per 1000 tonne-years).

This degree-year concept is similar to the idea of cumulative radiative forcing that has been identified by previous analyses as a climate indicator for tonne-year accounting[25]. Importantly however, the relationship between tonne-years and degree-years is grounded in the well understood physical relationship between cumulative emissions and temperature change, and can therefore be quantified linearly using the value of the TCRE. Avoided degree-years are also relevant for reducing climate impacts that have a strong inertial component to them, such as sea-level rise and permafrost melt. However, impacts that respond quickly to the amount of global temperature increase would be affected by avoided degree-years only during the time that degree-years continue to accumulate. And as with tonne-years, avoided degree years would only decrease peak warming if they continue to accumulate beyond the point in time that temperatures peak and begin to decline in response to successful fossil fuel $CO_2$ mitigation.

The equations presented above can also be used to understand the relationship between the rate of increase of tonne-years and the temperature benefit simulated by our climate model. The differentiated version of Eq. (4) can be written as:

$$\Delta T_R(t) = TCRE*TY'(t) \qquad (5)$$

where TY'(t) is the rate of increase of tonne-years over time. This equation clearly explains the four cases shown in Fig. 2a, b: case 1 represents an increasing climate benefit over time resulting from an increasing TY'(t); case 2 represents a sustained climate benefit resulting from constant *TY'(t)*; case 3 represents a decreasing TY'(t) leading to the loss of previously accrued temperature benefit; and case 4 represents the loss of all previously accrued temperature benefit which occurs at the time that TY'(t) reaches zero.

It is worth noting that our simulations represent only the $CO_2$ effect of land carbon storage, and not any associated biophysical effects on surface albedo, evapotranspiration or cloud cover that would also result from avoided deforestation or reforestation efforts[14,31,32]. Previous analyses have shown that the biophysical effects of land-use change also scale proportionately with land-use cumulative $CO_2$ emissions[33]; consequently we do not expect that these biophysical effects would alter the overall proportionality of tonne-years and degree-years that we have found here, nor the relationship between temperature benefit and the rate of increase of tonne-years. However, including potential biophysical effects would alter the slope of the tonne-year : degree-year relationship. In this case, rather than the TCRE itself, an "effective TCRE[34]" for land-use change $CO_2$ emissions that reflects the net $CO_2$ + biophysical effect on climate could be used to estimate the climate benefit of tonne-years of carbon storage. This would be especially important to consider in the case of afforestation projects in areas of seasonal snow cover, where the impact of surface albedo change can be large enough to negate (or even supersede) the climate benefit of enhanced carbon storage[8,35].

## Equivalency of temporary to long-term storage?
The concept of tonne-year accounting has been in the literature for several decades[24,25,36]. However, proponents of tonne-years have

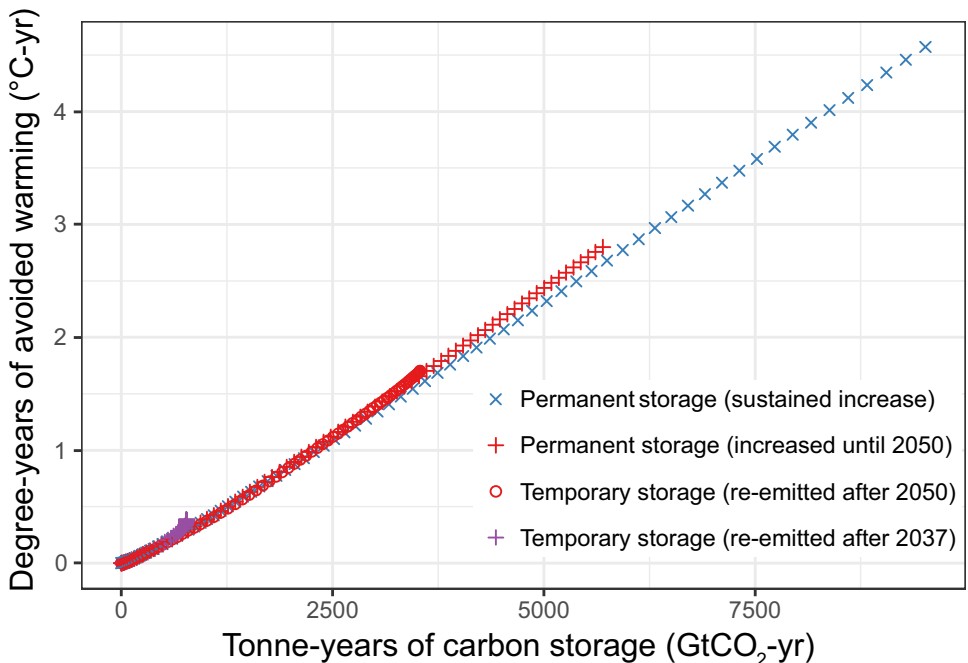

**Fig. 3 | Tonne-years of carbon storage are proportional to degree-years of avoided warming.** The relationship between total accumulated tonne-years and avoided degree-years of warming is an extension of the proportionality of cumulative $CO_2$ emissions and global temperature change, whereby the slope of the tonne-year : degree-year relationship is equal to the transient climate response to cumulative $CO_2$ emissions (TCRE). All four simulations presented in Figs 1 and 2 follow the same trajectory, as seen by the overlapping symbols in the lower left section of the plot. Source data are provided as a Source data file.

generally focused on the question of whether some amount of temporary carbon storage can be considered to be equivalent to some smaller amount of permanent carbon storage[26]. A range of equivalency factors have been proposed in an attempt to quantify how many tonne-years of temporary carbon storage would be needed to achieve an equivalent climate outcome as 1 tonne of permanent carbon storage[24–26]. These equivalency factors are typically constructed using some form of economic cost-benefit analysis, such that the cost of a unit emission of $CO_2$ is compared to the benefit of delaying that emission by some amount of time. The cost is generally quantified using an estimate of the atmospheric $CO_2$ response to an emissions, and some analyses further assume an economic discount rate which is used to decrease the present-day cost of a future emission[26]. The resulting equivalency factors range from about 30 to 130, and are claimed to represent the number of tonne-years of temporary storage that would be appropriate to offset the effect of one unit of $CO_2$ emissions.

Here we show that using an equivalency factor to infer the climate benefit of temporary carbon storage produces a time series of presumed temperature benefit that bears almost no resemblance to the actual avoided warming that results from temporary storage (Fig. 4). Using the equivalency factors from refs. 25,26 (128 and 31 tonne-years per tonne of permanent storage, respectively), we calculate the inferred temperature effect by first dividing the modelled tonne-years by an equivalency factor to produce the equivalent number of tonnes of permanent storage. Then, given that permanent storage (i.e., permanently avoided emissions) is proportional to avoided temperature change, we can use Eq. (2) to calculate the implied avoided warming associated with this amount of permanent storage. As shown in Fig. 4, the inferred avoided warming increases with the accumulation of tonne-years, with a smaller equivalency factor[26] producing a larger implied temperature benefit. Compared with the actual avoided warming over time (solid lines in Fig. 4, calculated here using Eq. (1)), the implied avoided warming calculated using tonne-year equivalency leads to an underestimate of near-term climate benefit, such that actual avoided warming during the time that carbon is being stored is larger than what is implied from the equivalency factor. Conversely,

after stored carbon has been re-released to the atmosphere, the amount of avoided warming decreases to zero, whereas the tonne-year equivalency factors incorrectly suggest an increasing and ultimately sustained climate benefit.

### Reimagining tonne-years as a complement to fossil fuel emissions reductions

Our results show that tonne-years of temporary land carbon storage, whether achieved via temporarily avoided emissions or via temporary removal efforts, have a well-defined climate effect that emerges from the proportionality of avoided cumulative emissions and avoided temperature increase. The proportionality constant of this relationship is the same as that which determines the climate response to cumulative $CO_2$ emissions (the TCRE). Consequently, the climate effect of tonne-years in a temporary carbon storage scenario can be quantified using the TCRE as an equivalent number of avoided degree-years. Furthermore, the amount of avoided warming over time can be related to the rate of increase of tonne-years of temporary storage. As long as the total number of tonne-years accumulate at a rate that is constant or increasing, the temperature difference from the baseline case will be sustained or will increase with time. However, as the rate of increase of tonne-years decreases towards zero, the temperature difference will be eroded and will return to the baseline temperature at the time that tonne-years stop accumulating. Consequently, for temporary tonne-years to affect peak temperature, the rate of accumulation of total tonnes years must be sustained until temperatures peak and begin to decline in response to fossil fuel $CO_2$ emissions decreasing to net zero.

Our results show also that the use of a single equivalency factor to relate tonne-years of temporary storage to an equivalent amount of permanent storage does not adequately capture the climate response to a temporary carbon storage scenario. Consequently, if such an equivalency factor is used to justify the offsetting of fossil fuel emissions with tonne-years of temporary storage, the immediate climate effect would be to slow near-term warming (during the time that carbon remains stored). However, if the stored carbon is subsequently lost to the atmosphere, the combined effect of this re-emission with

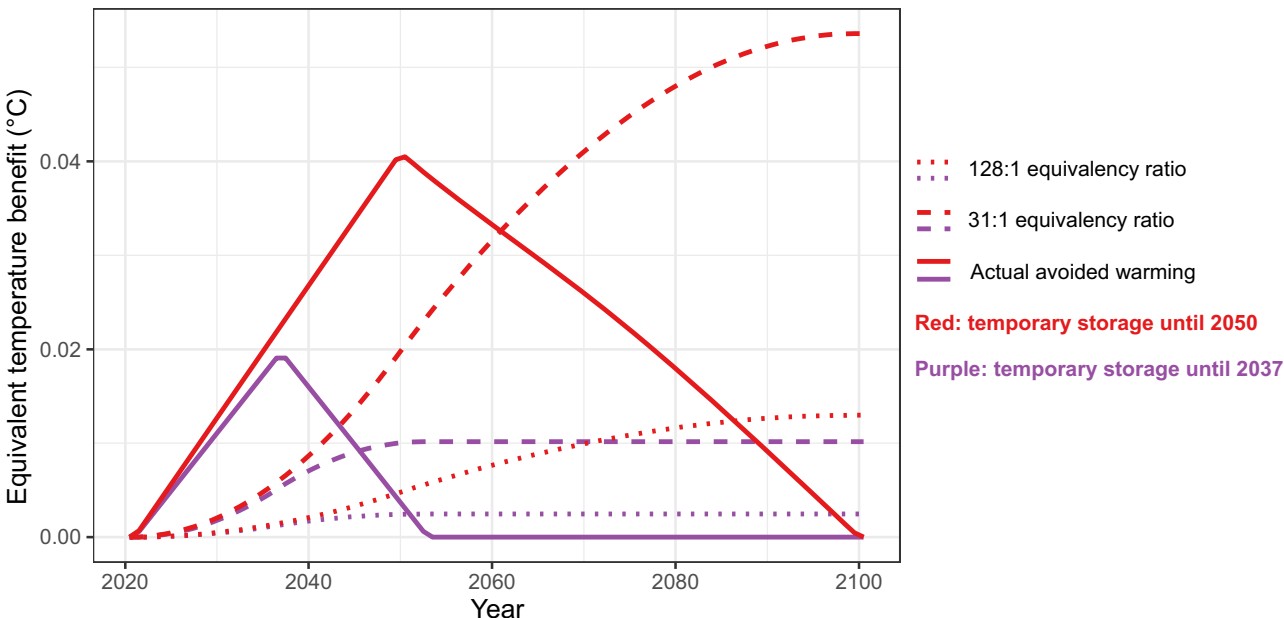

**Fig. 4 | Climate benefit of temporary carbon storage.** Climate benefit (avoided warming) as implied by equivalency factors that equate tonne-years of temporary storage to equivalent amounts of permanent storage (dashed and dotted lines), compared to the actual climate benefit that occurs in response to temporary carbon storage (solid lines). The use of equivalency factors to infer climate benefit leads to an underestimate of the near-term value of temporary storage, and a substantial overestimate of the long-term value of temporary storage. Source data are provided as a Source data file.

that of the previously offset (but emitted) fossil fuel $CO_2$ would lead to increased long-term warming. This trade-off of near-term avoided warming for additional long-term warming could create an impression of near-term mitigation progress, at the cost of compromising our ability to remain below long-term temperature targets.

In addition to this problematic trade-off associated with long-term warming, tonne-year equivalency factors also underestimate the near-term climate benefit of temporary storage. As a result, the use of these equivalency factors misrepresents the real and quantifiable climate benefit of temporary carbon storage that has the potential to make an important contribution to global climate mitigation efforts. This is especially the case if the time duration of temporary storage extends beyond the point of peak warming, in which case temporary storage can decrease the magnitude of the temperature peak[14]. Our results therefore suggest that temporary carbon storage has climate value, but its value is different from and not interchangeable with that of permanent storage or avoided fossil fuel $CO_2$ emissions.

Tonne-years accounting could therefore be used effectively to track the total amount of carbon storage that is achieved over time, and to correctly infer the resulting near-term climate benefit of that storage. Furthermore, because our proposed tonne-year approach is based on the instantaneous rate of change, quantifying the climate benefit can be done without prior knowledge of the longevity of the stored carbon. This removes the need to demonstrate in advance that the carbon will remain stored for a particular amount of time, which addresses a key barrier in establishing the climate value of a particular nature-based carbon storage effort. We have shown this for the specific case of land carbon storage achieved via avoided deforestation or reforestation efforts. However, in principle, a tonne-year accounting framework could be applied to track the climate benefit of any nature-based climate solution that is aimed at increasing carbon storage in non-permanent land or ocean reservoirs. If the goal was not to offset fossil fuel emissions, but rather to achieve a sustained global increase in the total number of tonne-years, this would result in a sustained climate benefit that is proportional to the rate of increase of tonne-years. This would represent an important contribution to global mitigation efforts that would complement fossil fuel emissions reductions.

A tonne-year accounting framework would additionally provide the flexibility to respond to losses of stored carbon that might occur either due to eventual deforestation, or as a result of natural disturbances. Given that the climate benefit results from the rate of accumulation of tonne-years achieved at all locations, the loss of stored carbon at one location could be compensated for by adding additional stored carbon at a different location. As long as the total number of tonne-years continues to increase at the same rate, the resulting climate benefit would be sustained over time.

This same principle could be applied at the level of an individual corporation or other institutional entity whose goal is to contribute to national or global climate mitigation efforts. An investment in either avoided emissions or carbon removal at a particular location would result in an amount of tonne-years that would increase each year according to the amount of carbon that remains stored. This would represent a sustained temperature benefit, which could be increased via the investment in an additional amount of stored carbon at a different location. If the carbon at one location is lost (via deforestation or other disturbance), then the climate benefit could be maintained via the investment in an equivalent additional amount of carbon storage at a different location. As long as the total number of tonne-years continues to increase, the climate benefit of this carbon storage investment would be sustained at a level that is proportional to the rate of tonne-year increase.

If reimagined in the way we have outlined here, tonne-year accounting could provide a critical contribution to climate mitigation efforts that would not depend explicitly on whether carbon remains stored permanently or only for some short amount of time. As a complement to (rather than as an offset for) fossil fuel emission reductions, the accumulation of tonne-years of temporary carbon storage would represent a real and quantifiable contribution to climate mitigation efforts which would not require any subjective choice of time-horizon nor guarantee of storage permanence. Regardless of the duration of the temporary carbon storage, the accumulated degree-years of avoided warming would contribute to reducing slow-responding climate impacts such as sea-level rise and permafrost melt. Furthermore, a sustained global increase in the number of tonne-years would result in a sustained global temperature benefit. If maintained in parallel with efforts to achieve net zero fossil fuel emissions, a global accumulation of tonne-years of carbon storage would have an important effect on limiting the peak temperature change that would occur if net zero emissions are achieved.

## Methods
### Earth system climate model
We used the University of Victoria Earth System Climate Model (UVic ESCM), version 2.10[37], an intermediate-complexity global climate model that includes a dynamic representation of land and ocean carbon cycle processes[38–40]. The model has a spatial resolution of 1.8° latitude and 3.6° longitude, and includes a general circulation ocean model, coupled to a single-layer energy-moisture balance atmospheric model and dynamic-thermodynamic sea-ice model[41]. Land carbon and dynamic vegetation processes are represented via five plant functional types as well as permafrost carbon storage[38,42], and the ocean carbon cycle includes both physical and biological carbon cycling, as well as representation of the sedimentary carbon cycle[19]. This model has been used and validated extensively over the past two decades to look at research questions such as assessing the effect of historical land-use change on climate, estimating the magnitude of climate-carbon cycle feedbacks, and quantifying the role of terrestrial and oceanic carbon cycle process in the context of both past and future climate scenarios[43–47].

### Simulation design
We ran the UVic ESCM using prescribed $CO_2$ emissions from fossil fuels and land-use change, in addition to other natural (solar and volcanic) and anthropogenic (non-$CO_2$ greenhouse gases and aerosols) climate drivers for the period from 1850 to 2150. Fossil fuel $CO_2$ emissions and other forcings followed historical observations up to the year 2015, and then followed the SSP1-1.9 or SSP1-2.6 mitigation scenarios that lead to net-zero fossil fuel $CO_2$ emissions during the second half of this century[48]. Against these background scenarios, we prescribed land-use emissions from the SSP3-7.0 scenario to represent the baseline business-as-usual case for future land-use (i.e. ongoing deforestation and consequent net positive land-use $CO_2$ emissions throughout the twenty-first century). This combination of land-use $CO_2$ emissions from SSP3-7.0 and other $CO_2$ emissions and non-$CO_2$ forcings from SSP1-1.9 or SSP1-2.6 constitute the "baseline" scenarios as shown in Figs. 1 and 2.

To represent the effect of increased land carbon storage that could result from avoided deforestation, reforestation or afforestation, we decreased prescribed land-use $CO_2$ emissions relative to the baseline scenario. The prescribed decrease of 3 GtCO₂ per year is consistent with estimates of the cost-effective carbon storage potential of either reforestation or avoided deforestation[1]; we do not differentiate between these two mechanisms of carbon sequestration, since both would cause a decrease of net land-use emissions either by slowing the loss of existing land carbon (avoided deforestation) or by reversing the effect of past losses (reforestation). In the context of current global land-use emissions of approximately 3.5 GtCO₂ per year[49], avoided deforestation, reforestation and afforestation would all have a similar effect on net land-use emissions.

We used four representative land-use scenarios to simulate the climate response to temporary and permanent land carbon storage. In the two temporary carbon storage simulations, we decreased prescribed land-use emissions relative to the baseline beginning at the year 2022 and sustained until the year 2037 (15 years) or 2050 (28 years). After this period, the same amount of stored carbon was re-emitted to the atmosphere by increasing prescribed emissions relative to the baseline over a period of 15 or 50 years, respectively (dashed purple and red lines in Fig. 1a). For the two permanent carbon storage simulations, we decreased prescribed land-use emissions relative to the baseline until the year 2050 (solid red line in Fig. 1a) or until the end of the simulation (solid blue line in Fig. 1a); in the latter simulation, we did not allow land-use emissions to become negative so as to represent a scenario that could be achieved via either reforestation or avoided deforestation. In these cases, the stored carbon was not re-emitted to the atmosphere.

In all scenarios, the decreased or increased emissions relative to the baseline land-use emission scenario are meant to represent the aggregate global effect of all individual carbon storage efforts occurring at different locations. For example, an individual delayed deforestation project would contribute a portion of one year's decreased emissions in one of the temporary storage scenarios, and when deforestation eventually occurs at this location, this would similarly contribute to a subsequent year's emissions increase. In aggregate, all individual projects would lead to the sustained annual decrease and/or increase of emissions relative to the baseline scenario.

### Tonne-year calculation

A tonne-year is defined as an amount of carbon stored in a particular reservoir, such as on land or in the atmosphere, integrated over some period of time[24–26]. Here, we calculated tonne-years from the perspective of the land carbon reservoir, so as to quantify the amount of carbon that is stored in the land carbon pool over time, relative to the baseline case where that carbon is instead emitted to the atmosphere. This approach leads to a number of tonne-years that is equal to the time-integrated difference in cumulative emissions between each land-use scenario and the baseline scenario (Fig. 1b). Rather than using tonne-years to represent the atmospheric consequence of an emission[25,26], we instead use tonne-years to quantify the cumulative stored land carbon over time. This land-based approach allows us to relate tonne-years of stored carbon to temperature change using the TCRE metric (the Transient Climate Response to Cumulative $CO_2$ Emissions) which represents the linear proportionality between cumulative emissions and global temperature change[21–23]. Similarly, we can derive a relationship between tonne-years of storage and avoided degree-years of temperature increase, where the slope of this relationship (shown in Fig. 3) is equal to the TCRE value of the UVic ESCM (approximately 0.45 °C per 1000 Gt $CO_2$)[37].

To calculate the temperature effect that would be implied using tonne-year equivalency factors (Fig. 4) we also made use of the TCRE relationship. We first calculated the implied equivalent amount of permanent storage by dividing our simulated tonne-years of temporary storage by equivalency factors of 31[26] or 128[25]. We then multiplied this implied permanent storage by the TCRE value (0.45 °C per 1000 Gt$CO_2$ of storage) to estimate the climate effect that would occur from this amount of permanent storage. This can be then compared with the actual climate effect of temporary storage, which we show in Fig. 4 as the cumulative emissions difference (Fig. 1b) multiplied by the TCRE.

### Data availability

Raw data plotted in the manuscript figures (which includes the input model data used for this study) are included as a supplementary Source data file. Source data are provided with this paper.

### Code availability

The model code for version 2.10 of the UVic ESCM is available on the official UVic ESCM webpage at http://terra.seos.uvic.ca/model/2.10.

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

## Acknowledgements
H.D.M and K.Z. acknowledge funding that supported this research from Microsoft and from the Natural Sciences and Engineering Research Council of Canada.

## Author contributions
H.D.M., K.Z., A.K. and A.L. participated in discussions that led to the conceptualisation and design of this study. A.K. carried out the model simulations and produced figure panels. H.D.M. led the writing of the manuscript and prepared the final figure versions. K.Z., A,K. and A.L. contributed to the writing, editing and review of the final manuscript.

## Competing interests
The authors declare no competing interests.
