## [Peer Review File · Nature Communications]

Accounting for the climate benefit of temporary carbon storage in natureEditorial Note: This manuscript has been previously reviewed at another journal that is not operating a transparent peer review scheme. This document only contains reviewer comments and rebuttal letters for versions considered at Nature Communications.

REVIEWER COMMENTS

Reviewer #1 (Remarks to the Author):

The authors addressed satisfactorily the few issues I had the first time I reviewed their manuscript. I recommend publication as is.

Reviewer #2 (Remarks to the Author):

The issue of the temporary nature of land carbon storage is important for climate mitigation policy. Currently, the accounting for this temporary storage is inadequate and results in inappropriate reporting. Hence, this paper about addressing the accounting for temporary storage is highly useful.

The recommendation of treating temporary storage in the land separate from, and complementary to, permanent storage or fossil fuel emissions, and that they are not interchangeable, is a useful contribution that builds on earlier recommendations for comprehensive carbon accounting that differentiates geocarbon and biocarbon in papers that address the broader issues of the accounting required for NbCS.

The last section on tonne-years as a complement to fossil fuel emissions reductions explains very well the difference in the climate benefit between temporary and permanent carbon storage: both provide benefits, but they are different and should not be treated as an offset.

However, a description of temporary versus permanent carbon storage would help to set the context of the accounting issue being addressed in the paper. This description would then be used as the basis for realistic examples of the types of land management mitigation activities that are being simulated.

A critical point is that it is only a small fraction of the land carbon stock that is subject to loss and gain due to disturbances and thus may be considered temporary; it is not the whole carbon stock of an ecosystem. This point is not explained, and the narrative appears to imply that it is the whole carbon stock that is temporary, which diminishes the perceived value of land and ecosystems for carbon storage. The activities that may be considered temporary include planting trees and then harvesting and combusting the biomass or delayed deforestation. Avoided deforestation, avoided harvesting, restoration, afforestation and soil carbon accumulation activities produce permanent carbon storage (with the exception of the small proportion that may be lost and gained due to disturbances although even these fluctuations can be averaged over the landscape scale).

Adopting the recommendation of the previous reviewer #2 to explain the concept of tonne-years using simple mathematical equations would enhance the paper, contributing particularly to explaining the concept to a general scientific audience. The concept is applicable at many levels and to specific types of activities, and does not necessarily need outputs from a climate model to be useful. The introduction of

the baseline emissions following a SSP pathway of positive but decreasing emissions, combined with simulation of the global aggregation of all mitigation activities, can be presented as a second stage. The additional information that is provided by these model simulations could then be explained.

L75-78 This is a key point for determining the efficacy of accounting for tonne-years of temporary storage in terms of climate response. Although the statement is referenced, it would be useful to state the problem explicitly so that this former approach can be differentiated from the proposed new approach. The earlier approach referred to as “having little bearing” and “not grounded in climate science”.

The difference in approach to the use of tonne-years appears to be that the previous use of tonne-years was as an economic parameter involving cost benefit analysis, whereas the approach proposed in this paper is to use tonne-years as purely a physical parameter. The calculation of this physical parameter is not changed. The difference in approach is in how the physical parameter is then used. I think this could be explained in the Introduction and would help to clarify the ‘new approach’ presented in this paper.

Additional description of temporary carbon storage in the land sector is needed. An impression arising from the current paper is that all / or much of the carbon storage in the land sector is temporary – but this is not the case. An ecological perspective of carbon storage involves the landscape approach of dynamics in carbon stocks over space and time in response to disturbances, but within the bounds of a retained average stock. A mitigation policy to avoid deforestation can represent a permanent change that avoids the emissions from most of the carbon stock in the forest ecosystem. The only emissions that may occur in the future are due to natural disturbances which only release a small proportion of the carbon stock that is then regained through regeneration (unless a complete change in ecosystem type is triggered by the disturbance).

Simulations of temporary carbon storage that decrease land emissions only until 2037 or 2050 represent only limited scenarios in the land sector, such as extending harvest rotation lengths or delayed (not avoided) deforestation. However, many mitigation scenarios represent long-term indefinite reductions in emissions and increases in removals, such as avoided deforestation, restoration and afforestation. Recognising that the scenarios presented are theoretical, it would greatly aid interpretation of the results if actual land management practices were described that represent each scenario. The scenario of temporary carbon storage - where an amount of carbon is sequestered / or emissions avoided, and then after a period all the carbon is emitted – appears to represent biomass that is grown and then harvested and combusted. The proposition of delayed deforestation is not a forest management strategy that should be advocated for climate mitigation or a sustainable environment. The scenario of temporary carbon storage does not appear to relate to mitigation in the land sector, such as avoided deforestation, reforestation and afforestation.

Figure 1a.

- The lines on the graph are not clear. There appear to be dashed lines behind solid lines. Where there is a need for two lines, it would be clearer to position them next to each other, so that the trajectory of both scenarios can be seen. This is the case in figure 2 as well.

- The emissions from the temporary storage that occur in 2037 and 2050 appear that all the stored carbon is emitted in 1 year (the lines are vertical), which may occur for the scenario of deforestation

(although in reality a large area could not be cleared in 1 year). However, it is unclear why the high rate of emissions continues for 15 or 50 years. A more practical scenario would be that emissions increase gradually over time as more areas are cleared each year (after the policy of avoided deforestation was rescinded).

- It is unclear why there is a change in direction of the solid lines at 2050, and what the red line from 2100 to 2120 represents.

Figure 3. Please state in the figure caption whether each of the curves are following the same trajectory. Understandably it is difficult to show each of the symbols on top of each other.

Figure 4. It is unclear what is represented by the black solid line.

In the last section about application of the concept of tonne-years for mitigation policy, it appears that determining the climate benefit from tonne-years is dependent on knowing the length of time of the temporary storage, i.e. when it will be emitted in the future. In terms of land management policy, this time frame is often not known, or not ensured. Cases where the time frame may be known are harvest rotations of forests or management agreements for delaying deforestation. Some discussion about these practicalities of implementing the concept for climate mitigation policy would be helpful.

REVIEWER COMMENTS

Reviewer #1 (Remarks to the Author):

The authors addressed satisfactorily the few issues I had the first time I reviewed their manuscript. I recommend publication as is.

Thank you again for your work and time spent reviewing our paper.

Reviewer #2 (Remarks to the Author):

The issue of the temporary nature of land carbon storage is important for climate mitigation policy. Currently, the accounting for this temporary storage is inadequate and results in inappropriate reporting. Hence, this paper about addressing the accounting for temporary storage is highly useful.

The recommendation of treating temporary storage in the land separate from, and complementary to, permanent storage or fossil fuel emissions, and that they are not interchangeable, is a useful contribution that builds on earlier recommendations for comprehensive carbon accounting that differentiates geocarbon and biocarbon in papers that address the broader issues of the accounting required for NbCS.

The last section on tonne-years as a complement to fossil fuel emissions reductions explains very well the difference in the climate benefit between temporary and permanent carbon storage: both provide benefits, but they are different and should not be treated as an offset.

Thank you for your work and time spent reviewing our paper, and for the helpful suggestions and comments that you have made below. Thank you also for the links to the two previous papers noted above, which we have now cited in the opening text.

However, a description of temporary versus permanent carbon storage would help to set the context of the accounting issue being addressed in the paper. This description would then be used as the basis for realistic examples of the types of land management mitigation activities that are being simulated. A critical point is that it is only a small fraction of the land carbon stock that is subject to loss and gain due to disturbances and thus may be considered temporary; it is not the whole carbon stock of an ecosystem. This point is not explained, and the narrative appears to imply that it is the whole carbon stock that is temporary, which diminishes the perceived value of land and ecosystems for carbon storage. The activities that may be considered temporary include planting trees and then harvesting and combusting the biomass or delayed deforestation. Avoided deforestation, avoided harvesting, restoration, afforestation and soil carbon accumulation activities produce permanent carbon storage (with the exception of the small proportion that may be lost and gained due to disturbances although even these fluctuations can be averaged over the landscape scale).

Thank you for flagging this. We have added a new paragraph to the introduction that describes the general context of permanent vs temporary carbon storage in land ecosystems, with particular mention of reforestation, afforestation and avoided (or delayed) deforestation. We have also clarified that at large spatial and temporal scales, the effect of disturbances is generally small and also balanced by regrowth. However, given that any particular forest-based carbon storage effort is vulnerable to

disturbances, this becomes a critical uncertainty in the context of carbon accounting efforts; here the permanence of particular plots of carbon is not guaranteed and is a key concern, especially when used to offset fossil-fuel emissions. We hope this revised introduction helps to set the context for our analysis and subsequent conclusions.

Adopting the recommendation of the previous reviewer #2 to explain the concept of tonne-years using simple mathematical equations would enhance the paper, contributing particularly to explaining the concept to a general scientific audience. The concept is applicable at many levels and to specific types of activities, and does not necessarily need outputs from a climate model to be useful. The introduction of the baseline emissions following a SSP pathway of positive but decreasing emissions, combined with simulation of the global aggregation of all mitigation activities, can be presented as a second stage. The additional information that is provided by these model simulations could then be explained.

As suggested, we have now included a version of equations suggestion by the previous reviewer #2 in the section titled "What is the climate equivalent of a tonne year". We agree that these equations help to explain in simple terms the model simulations shown in the previous figures. Furthermore, we actually used the first two of these equations to generate the lines plotted in Figure 4, and so we are now able to reference this explicitly which helps to clarify this figure also.

L75-78 This is a key point for determining the efficacy of accounting for tonne-years of temporary storage in terms of climate response. Although the statement is referenced, it would be useful to state the problem explicitly so that this former approach can be differentiated from the proposed new approach. The earlier approach referred to as "having little bearing" and "not grounded in climate science".

We have added a new sentence here that summarizes why past uses of tonne-year accounting are not well grounded on climate science: "Previous analyses have focussed only on the atmospheric CO2 response to a pulse CO2 emission, and have not quantified the temperature response to this atmospheric CO2 change, nor considered the difference between short- and long-term climate responses."

The difference in approach to the use of tonne-years appears to be that the previous use of tonne-years was as an economic parameter involving cost benefit analysis, whereas the approach proposed in this paper is to use tonne-years as purely a physical parameter. The calculation of this physical parameter is not changed. The difference in approach is in how the physical parameter is then used. I think this could be explained in the Introduction and would help to clarify the 'new approach' presented in this paper.

Thanks for this suggestion – we have added a clarifying sentence: "We propose that tonne-years, used only as a physical metric of carbon storage over time rather than an economic metric of equivalency, could be used effectively to estimate the climate response to nature-based carbon storage."

Additional description of temporary carbon storage in the land sector is needed. An impression arising from the current paper is that all / or much of the carbon storage in the land sector is temporary – but this is not the case. An ecological perspective of carbon storage involves the landscape approach of dynamics in carbon stocks over space and time in response to disturbances, but within the bounds of a retained average stock. A mitigation policy to avoid deforestation can represent a permanent change that avoids the emissions from most of the carbon stock in the forest ecosystem. The only emissions that may occur in the future are due to natural disturbances which only release a small proportion of the

carbon stock that is then regained through regeneration (unless a complete change in ecosystem type is triggered by the disturbance).

As noted above, we have added new text to the introduction that describes the broader ecological context of land carbon cycling, before focussing on the particular problems that arise in the context of using land carbon as part of carbon accounting efforts.

Simulations of temporary carbon storage that decrease land emissions only until 2037 or 2050 represent only limited scenarios in the land sector, such as extending harvest rotation lengths or delayed (not avoided) deforestation. However, many mitigation scenarios represent long-term indefinite reductions in emissions and increases in removals, such as avoided deforestation, restoration and afforestation.

We agree that most future scenarios include the assumption that land-based mitigation efforts can lead to indefinite emission reductions and often net carbon removals that persist for decades into the future. These assumptions can also be critiqued, this is of course not the focus of our paper. Our particular scenarios are not meant to represent any particular exiting future scenario, but rather to explore the consequences of temporary vs. permanent storage when land-based mitigation efforts are applied as an addition to a baseline scenario.

Recognising that the scenarios presented are theoretical, it would greatly aid interpretation of the results if actual land management practices were described that represent each scenario.

This information was included already in the methods section, though we agree that more details in the main text would be helpful here. The text that we have now added to the introduction includes explicit reference to the general types of land management practices (i.e. reforestation, afforestation, avoided deforestation and delayed deforestation) that are relevant to the scenarios that we developed for this analysis

The scenario of temporary carbon storage - where an amount of carbon is sequestered / or emissions avoided, and then after a period all the carbon is emitted – appears to represent biomass that is grown and then harvested and combusted.

This is correct, but this scenario could also represent biomass that is grown and subsequently lost to natural disturbance. We have added the following sentence here to clarify: “These temporary storage simulations could reflect scenarios whereby deforestation activities are delayed rather than avoided permanently, or whereby sequestered carbon is subsequently lost as a result of climate-driven increases in natural disturbances.”

The proposition of delayed deforestation is not a forest management strategy that should be advocated for climate mitigation or a sustainable environment.

We agree. However delayed deforestation is currently being advocated for in carbon offset markets, using economic-based tonne-year arguments to establish its climate value. This is highly problematic and is one of the motivations for our analysis.

The scenario of temporary carbon storage does not appear to relate to mitigation in the land sector, such as avoided deforestation, reforestation and afforestation.

We agree that over large spatial and temporal scales, the question of whether land carbon remains stored permanently or temporarily is not as relevant. However, in the context of carbon accounting, this is a key problem that is highly relevant. All of these forest management strategies, when accounted for at the scale of an individual forest plot, are subject to potential loss of stored carbon, and hence could result in temporary carbon storage. Our revised introduction should now clarify this key problem.

Figure 1a.

- The lines on the graph are not clear. There appear to be dashed lines behind solid lines. Where there is a need for two lines, it would be clearer to position them next to each other, so that the trajectory of both scenarios can be seen. This is the case in figure 2 as well.

OK we have offset the lines slightly for improved clarity and have noted this in the figure caption.

- The emissions from the temporary storage that occur in 2037 and 2050 appear that all the stored carbon is emitted in 1 year (the lines are vertical), which may occur for the scenario of deforestation (although in reality a large area could not be cleared in 1 year). However, it is unclear why the high rate of emissions continues for 15 or 50 years. A more practical scenario would be that emissions increase gradually over time as more areas are cleared each year (after the policy of avoided deforestation was rescinded).

These are annual emissions, so all stored carbon is not emitted in 1 year; rather in this year, the previously stored carbon begins to be re-emitted in that year. We have clarified this in the figure caption and main text by noting that the abrupt increase in annual emissions represents a switch from gradual accumulation of stored carbon (emissions below baseline) to gradual re-emission (emissions above baseline) and that this re-emission period occurs for either 15 or 50 years.

- It is unclear why there is a change in direction of the solid lines at 2050, and what the red line from 2100 to 2120 represents.

We hope that the revised figure with offset lines helps to clarify this, since the full trajectory of all five simulations is now more clearly visible.

Figure 3. Please state in the figure caption whether each of the curves are following the same trajectory. Understandably it is difficult to show each of the symbols on top of each other.

OK we have clarified that the curves do follow the same trajectory, and that the symbols overlap in the lower left portion of the plot.

Figure 4. It is unclear what is represented by the black solid line.

Thank you for pointing this out – the black line here was meant to represent both coloured solid lines, but I can see why this might have been confusing. We have now revised the legend of this plot to show the meaning of each individual line more clearly.

In the last section about application of the concept of tonne-years for mitigation policy, it appears that determining the climate benefit from tonne-years is dependent on knowing the length of time of the temporary storage, i.e. when it will be emitted in the future. In terms of land management policy, this time frame is often not known, or not ensured. Cases where the time frame may be known are harvest

rotations of forests or management agreements for delaying deforestation. Some discussion about these practicalities of implementing the concept for climate mitigation policy would be helpful.

Thank you for highlighting the need for additional clarify in this section. We have added the following sentences to emphasize that knowing the length of time of storage a priori is not required in our proposed approach: "Furthermore, because our proposed tonne-year approach is based on the instantaneous rate of change, quantifying the climate benefit can be done without prior knowledge of the longevity of the stored carbon. This removes the need to demonstrate in advance that the carbon will remain stored for a particular amount of time, which removes a key barrier in establishing the climate value of a particular nature-based carbon storage effort."